# Assessment of the Hydrochemical Characteristics and Formation Mechanisms of Groundwater in A Typical Alluvial-Proluvial Plain in China: An Example from Western Yongqing County

**Xueshan Bai [1], Xizhao Tian [1], Junfeng Li [2], Xinzhou Wang [1,\*], Yi Li [1] and Yahong Zhou [3]**

[1]  Hebei Key Laboratory of Environment Monitoring and Protection of Geological Resources, Hebei Geological Environment Monitoring Institute, Shijiazhuang 050021, China; hjjcybaixs@163.com (X.B.); 13513399710@163.com (X.T.); 13933129093@163.com (Y.L.)

[2]  Hebei Institute of Hydrogeology and Engineering Geology, Shijiazhuang 050000, China; lijie7098@163.com

[3]  School of Water Resources and Environment, Hebei GEO University, Shijiazhuang 050031, China; zhyh327@163.com

\*  Correspondence: xzwang050031@163.com

**Abstract:** The geographic location of Yongqing County is optimal, covering the center of the Beijing, Tianjin, and Baoding triangle. However, the economic and social development of Yongqing County in recent years has resulted in negative impacts on groundwater. Therefore, investigating the current status of groundwater chemistry in Yongqing County is of great significance to provide a useful basis for future studies on groundwater quality assessment. The aim of this study is to assess the hydrochemical characteristics and formation mechanisms of the unconfined aquifers of Yongqing County using descriptive statistical and multivariate statistical methods. In addition, ionic ratios, Piper diagram, Gibbs diagrams, and PHREEQC software were used in this study to determine the main factors influencing the hydrochemical characteristics of the unconfined aquifers. The results suggested slightly alkaline groundwater of the unconfined aquifers in the western part of Yongqing County, belonging to the fresh-brackish groundwater type. In addition, the hydrochemistry facies types in the study area are complex, consisting of four facies types, namely $HCO_3^--Mg\cdot Ca$, $HCO_3^--Na$, $HCO_3^--Na\cdot Ca$, and $HCO_3^--Na\cdot Mg$. On the other hand, the main factors influencing the hydrochemical characteristics of groundwater are mineral dissolution followed by some anthropogenic pollution. Rock dominance was the main influencing factor, demonstrated by the dissolution of silicate and carbonate rock minerals. In addition, the alternating adsorption of cations occurring in the aquifer plays a non-negligible effect on the hydrochemical characteristics of the unconfined aquifers in the study area. In fact, the validation results using PHREEQC inverse hydrogeochemical simulations demonstrated consistent conclusions with those mentioned above. According to the findings obtained, the dissolution of carbonate and silicate minerals as well as $Na^+$, $K^+$, and $Ca^{2+}$ ion exchange in the aquifer are the main factors influencing the hydrochemical characteristics of the unconfined aquifers of Yongqing County. The recommendations suggest put forward in this research are helpful to understand the formation mechanism of hydrochemistry in typical alluvial proluvial plain and provide insights for decision makers to protect the groundwater resources.

**Keywords:** groundwater quality; water chemistry characteristics; multivariate statistical analysis; PHREEQC; typical mountain front tilted plain

## 1. Introduction

Water and environmental issues have attracted increasing attention over the past few decades due to the development of the economy and society. In fact, although the total amount of water resources in China is high, the per capita water use is less than

a quarter of the world's per capita water use [1]. Therefore, China has been listed by the United Nations among the 13 water-poor countries. In the current situation of water scarcity in China, groundwater resource is becoming particularly important as they are considered the main source of water supply for most cities. To assess the current status of the regional groundwater environment, it is crucial to conduct studies on the hydrochemical characteristics and formation mechanisms of groundwater. These studies can, indeed, provide a general understanding of the hydrochemical characteristics of aquifers and the main factors influencing groundwater for future research.

At present, most studies on groundwater have been focused on hydrochemical characteristics and formation mechanisms of groundwater, hydrogeochemical simulation [2], human health risk assessment associated with organic and mineral groundwater pollution, groundwater pollution remediation technologies, and source of groundwater recharge and pollution using isotope technology. For example, Zhao et al. [3] assessed the spatiotemporal distribution of hydrochemical characteristics and formation mechanisms as well as the renewal ability of a confined aquifer in Hangzhou Bay New Area and showed that the aquifer was formed in the Late Pleistocene, with the absence of any hydraulic connection between this aquifer and other aquifers, which suggests a low regeneration capacity. Sang et al. [4] used the PHREEQC software to assess the hydrogeochemical characteristics of confined and unconfined aquifers in the delta area of the Nakdong River basin, Busan, Korea, which is located in the southeast of Beijing, China, and revealed unsaturated salt and supersaturated dolomite and calcite indices, which indicates that dissolution of carbonate rocks and ion exchange of major ions are the main hydrogeochemical processes in groundwater. Liu et al. assessed the groundwater quality and human health risk of groundwater samples collected from Yulin City and revealed good groundwater quality in the study area, with the presence of nitrate ($NO_3^-$) contamination in groundwater in agricultural areas [5], which suggests that reasonable groundwater management strategy should be established. Propp et al. [6] assessed groundwater quality in 20 historic landfills in Ontario, Canada, and indicated that most groundwater was strongly influenced by waste leachate, as landfills are long-term sources of several types of contaminants in groundwater. On the other hand, Pham et al. developed a new technique for the remediation of persistent contaminants, such as dense nonaqueous phase liquids in groundwater [7]. Moreover, innovative methods involving the sustained release of a selected reagent, namely persulfate, through pellets made from inorganic materials (e.g., zeolite, diatomite, and silica flour) are reported through a proof-of-concept study. This study demonstrated the potential feasibility of sustained persulfate release from inert matrices for groundwater treatment. Phan et al. used isotopic tracing techniques, namely $\delta^2H$, $\delta^{18}O$, $\delta^{13}C$, $\delta^3H$, and $\delta^{14}C$ activities, to assess the groundwater recharge, runoff, and discharge conditions of groundwater in the High Plains region of northeastern New Mexico, USA, which suggests that groundwater in the study area may be a mixture of Holocene groundwater and modern water co-existence [8]. Surface features (e.g., alluvial channels) promote the groundwater recharge, resulting in higher recharge rates in the region than the regional average rates.

This study aims to assess the regional hydrochemical characteristics of groundwater in a typical pre-hill alluvial plain area [9]. In addition, an inverse simulation technique was performed using the PHREEQC software to achieve qualitative and quantitative analyses of regional groundwater chemistry in the study area [10]. In this study, the chemical characteristics and formation mechanism of groundwater in the phreatic aquifer in a typical pre-hill tilted plain area were studied [11,12]. The study area is located in the eastern plain area of the Taihang Mountains, which is a typical pre-mountain sloping plain. The study area is a part of a typical alluvial-proluvial plain, characterized by a single groundwater facies type and good groundwater quality. However, the increasingly frequent human activities in recent years have affected quantitatively and qualitatively the groundwater in the study area. This study provides a solid theoretical basis for ensuring the safety of drinking water in Yongqing County and achieving the sustainable use of water resources.

## 2. Study Area Overview

### 2.1. Physical Geography Overview

Yongqing County is a county-level city in Langfang (116°22′–116°43′ E and 39°07′–39°28′ N), covering the middle of Hebei Province, the hinterland of the North China Plain, and the middle section of the Beijing-Tianjin Golden Corridor, with a total surface area of 776 km². The centers of the triangle of Beijing, Tianjin, and Baoding are located in the hinterland of Beijing and the economic circle around the Bohai Sea (Figure 1). The study area is located 60 km north of the capital Beijing, 60 km east of Tianjin City, 80 km from the capital airport, and 100 km from Tianjin Xingang. The study area is mainly located in the western part of Yongqing County and the eastern plain area of the Taihang Mountains, covering four townships, namely Yongqing Township, Longhuzhuang Township, Houyi Township, and Liujie Township, covering a total area of about 374 km². The northwestern and southeastern parts of the study area are characterized by high and low topography, respectively. In addition, the study is not flat and has some irregularities due to the interactive deposition of rivers, with the presence of several lowlands and depressions distributed at river crossings. The rivers are characterized by low erodibility since they are plain-type. According to the type of geomorphogenesis and surface morphology, the study area is located in a secondary geomorphic unit of alluvial the microtilt plain area.

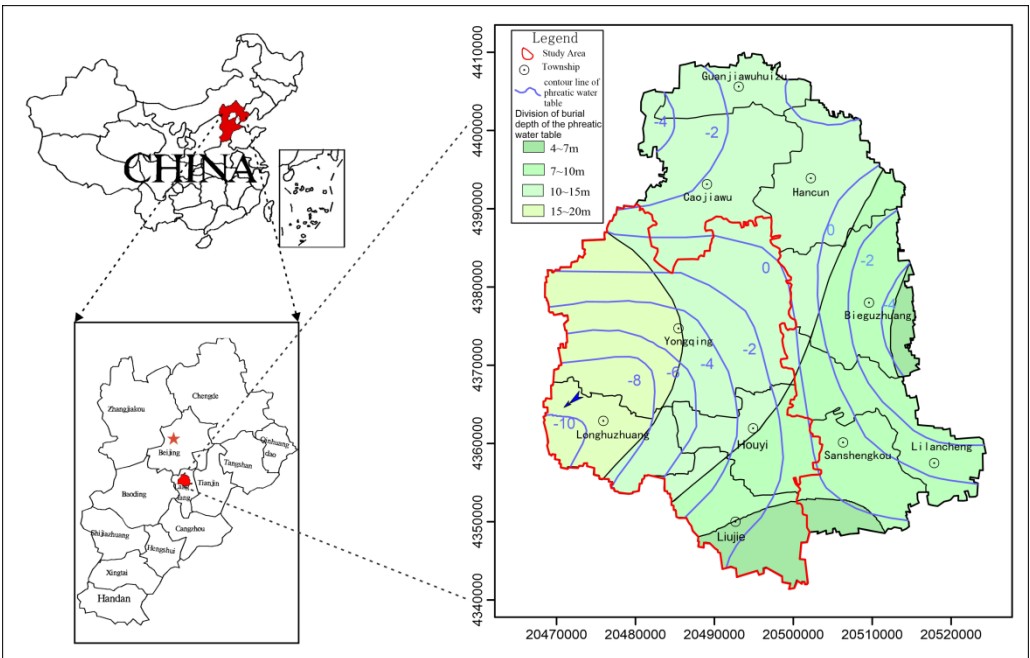

**Figure 1.** Groundwater burial depth classification and iso-water level map in the study area.

### 2.2. Socio-Economic Conditions

Yongqing is located in the new urban area of Langfang and a new space in Beijing, where the economy and society are developing rapidly. According to the statistics, the county's gross regional product was estimated at 22.01 billion yuan in 2020, while the general public budget revenue was estimated at 1.699 billion yuan. In addition, the per capita disposable income of urban and rural residents was 40,411 and 18,576 yuan, respectively. According to the data of the Seventh National Census Bulletin, released in November 2020, the resident population of Yongqing County is about 384,767 people. On the other hand, besides the largest gas-fired industrial zone in the northern part of Yongqing County, eight leading industries are present in Yongqing County, including chemical, pharmaceutical, cable, musical instrument, and building materials industries. Agriculture activities consist mainly of planting and animal husbandry. Yongqing has become a national pollution-free vegetable base county.

### 2.3. Hydrogeological Overview

The exposed strata in the study area consist of Cenozoic Quaternary loose deposits, with a thickness range of 350–500 m. The genetic types are complex, with a dominance of alluvial and lacustrine floodplains and their transitional types. However, the underlying strata of the Quaternary System consist of the Neoproterozoic and Paleocene strata.

The study area is located in the Yongding River area, including the alluvial-proluvial plain area and paleochannel zone. The groundwater system in the study area is classified into four aquifer groups, of which the first and second groups are shallow groundwater aquifers, consisting primarily of fine-to-medium sands and silty sand with gravel. The bottom boundary depth and aquifer thickness ranges of these aquifer groups are 160–180 m and 30–50 m, respectively. The third aquifer group is the deep groundwater aquifer, with bottom boundary depth and aquifer thickness ranges of 350–385 m and 60–100 m, respectively. The lithology of this aquifer changes from gravelly sand and medium sand to fine sand from north–south. The fourth aquifer group is the deep groundwater aquifer group, with bottom boundary depth and aquifer thickness ranges of 420–520 m and 20–40 m, respectively. There is no isolation factor among the aquifers, but the water flow is not quite the same in each layer because the aquifers are not homogeneous. The lithology of this aquifer consists mainly of fine-to-medium sands. On the other hand, the main recharge source of the shallow aquifers is atmospheric precipitation, followed by infiltration of irrigation and surface water and lateral runoff. Whereas mining activities are the main discharge source of the shallow aquifers, followed by runoff discharge downstream. The general shallow groundwater runoff trend is Northwest to Southeast, with a hydraulic gradient range of 0.8–1.4%. However, due to the existence of a local groundwater funnel, the direction of groundwater flow in some areas has changed (Figure 1).

### 3. Materials and Methods

#### 3.1. Sample Collection and Analysis

In this study, a total of 14 groundwater samples were collected from shallow aquifer I+II during the monsoon period (June–August, 2011), with sampling buried depths of groundwater levels ranging from 50 to 150 m. The groundwater samples were collected appropriately according to the Code of Practice for Groundwater Environmental Monitoring (HJ 164-2020). The collected samples were first stored in a refrigerated box and then sent to the laboratory for chemical analysis. Samples requiring additives, added before sampling is completed. Finally, the status of the samples is checked regularly. The spatial distribution map of groundwater sampling points area was generated using the MAPGIS 6.7 software (Figure 2). It can be seen from Figure 2 that the sampling points were evenly distributed, covering the aquifer area in the study area.

The analytical data include the results of analyses of more than 20 parameters, in which the groundwater depth, longitude, latitude, smell and taste, turbidity, and naked-eye visible matter were measured and recorded in-situ. Other hydrochemical parameters, namely pH, hardness, total dissolved solids (TDS), mineralization, silicon dioxide ($SiO_2$), potassium ($K^+$), sodium ($Na^+$), calcium ($Ca^{2+}$), magnesium ($Mg^{2+}$), sulfate ($SO_4^{2-}$), chloride ($Cl^-$), bicarbonate ($HCO_3^-$), iron ($Fe^{2+}/Fe^{3+}$), fluorine ($F^-$), nitrite ($NO_2^-$), nitrate ($NO_3^-$), arsenic (As), and manganese (Mn). However, $Fe^{3+}$ was not considered in this study due to its low concentration in groundwater, which was below the detection limit.

All the analytical methods used in the analyses were carried out according to the standard methods reported by Nsabimana et al. [13]. On the other hand, in order to check the reliability of the water quality analysis results, the ionic balance was used calculated according to the following formula:

$$E(\%) = \frac{\sum N_c - \sum N_a}{\sum N_c + \sum N_a} \times 100 \tag{1}$$

where E is the relative error; $N_c$ is the concentration of the cation in the groundwater sample (meq/L); $N_a$ is the concentration of the anion (meq/L).

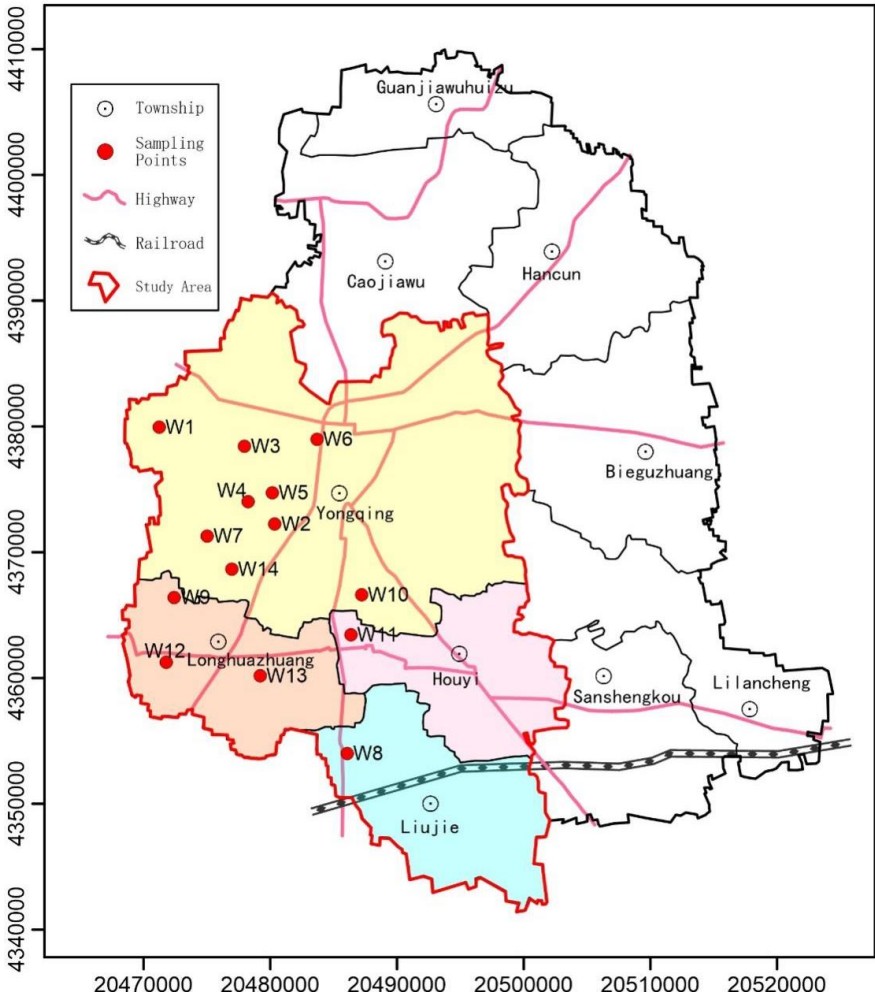

**Figure 2.** Spatial distribution of sampling points in the study area.

According to the results obtained, all the analytical data of groundwater samples showed E values less than ±5%, which suggests appropriate analytical methods.

### 3.2. Data Analysis Methods

In addition to, descriptive statistics, the principal component analysis test was performed in this study using the SPSS software to determine the main influencing factors affecting the hydrochemical characteristics of groundwater in Yongqing County [14,15]. In addition, ionic ratios were used to analyze the alternate adsorption of cations and investigate the main sources of hydrochemical elements in groundwater. On the other hand, the PHREEQC software was used for hydrogeochemical inversion simulations.

## 4. Results and Discussion

### 4.1. Descriptive Statistical Analysis

In order to determine the hydrochemical characteristics of groundwater, the SPSS statistical analysis software was used to perform descriptive statistical analysis (minimum, median, maximum, mean, and standard deviation) on the results data of 18 hydrochemical parameters of groundwater samples (Table 1).

According to the results obtained, pH values ranged from 7.5 to 8.3, with a mean value of 7.8, which indicates that the pH of groundwater in the western part of Yongqing County is slightly alkaline. In addition, the mineralization ranged from 345.5 to 1800.9 mg/L, with a mean value of 896.8 mg/L. In fact, some groundwater samples showed high mineralization values (above 1 g/L), indicating fresh-brackish groundwater. The TDS concentration

range was 243.7–1333.6 mg/L, with a mean value of 638.4 mg/L, showing significant spatial variation in the TDS concentrations in the study area with the presence of higher concentrations than that of the National Standard for Drinking Water (GB 5749-2006). On the other hand, $Na^+$ was the most abundant cation in groundwater, with a mean concentration value of 116.764 mg/L, followed, respectively, by $Ca^{2+}$, $Mg^{2+}$, $K^+$, and $Fe^{2+}$. Whereas $HCO_3^-$ was the highest anion in groundwater, with a mean concentration value of 516.77 mg/L, followed, respectively, by $SO_4^{2-}$, $Cl^-$, $NO_3^-$, $F^-$, and $NO_2^-$.

**Table 1.** Descriptive statistics of chemical parameters of groundwater.

| Item | Min(mg/L) | Med(mg/L) | Max(mg/L) | Mean(mg/L) | SD |
|---|---|---|---|---|---|
| TDS | 243.683 | 669.979 | 1333.622 | 638.442 | 331.674 |
| $HCO_3^-$ | 203.589 | 545.767 | 934.549 | 516.770 | 255.924 |
| TH | 28.912 | 358.710 | 840.592 | 328.933 | 224.884 |
| $Na^+$ | 44.100 | 106.100 | 220.700 | 116.764 | 66.109 |
| $SO_4^{2-}$ | 26.500 | 57.300 | 244.200 | 87.821 | 65.843 |
| $Ca^{2+}$ | 7.585 | 52.095 | 113.771 | 53.407 | 32.334 |
| $Cl^-$ | 9.227 | 46.490 | 170.345 | 50.546 | 42.496 |
| $Mg^{2+}$ | 2.421 | 54.967 | 143.836 | 47.487 | 38.019 |
| $SiO_2$ | 13.290 | 18.502 | 23.930 | 18.671 | 2.458 |
| pH | 7.500 | 7.770 | 8.310 | 7.806 | 0.239 |
| $NO_3^-$ | 1.420 | 3.780 | 9.590 | 3.687 | 2.077 |
| $F^-$ | 0.175 | 0.641 | 3.083 | 0.896 | 0.780 |
| $K^+$ | 0.400 | 0.600 | 2.600 | 0.754 | 0.573 |
| $Fe^{2+}$ | 0.025 | 0.175 | 1.110 | 0.283 | 0.340 |
| Mn | 0.005 | 0.171 | 0.477 | 0.164 | 0.161 |
| $NO_2^-$ | 0.002 | 0.006 | 0.064 | 0.014 | 0.017 |
| As | 0.001 | 0.002 | 0.004 | 0.002 | 0.001 |

By comparing the average concentration values of the hydrochemical parameters of groundwater with the Drinking Water Standards' (GB 5749-2006), it was found some ions exceeded the drinking limit values, which suggests a deterioration of the groundwater quality in the study area [16,17]. The main manifestations were $SO_4^{2-}$, $NO_3^-$, Mn, $F^-$, $Na^+$, and two combined indicators TH and TDS.

*4.2. Basic Characteristics of Groundwater Chemistry*

The groundwater facies types in the study area were determined by calculating first the relative content of anion and cation in the collected groundwater, and then drawn the Piper diagram (Figure 3). Actually, the groundwater facies type can be determined by plotting groundwater sample points in all three zones (2 triangles and 1 diamond) of the Piper diagram [18].

Through the comparison between the calculation and Piper diagram [19], it was found that among the 14 groundwater sample points, 2, 2, 3, and 7 sampling points fall in $HCO_3^-$−Mg·Ca, $HCO_3^-$−Na, $HCO_3^-$−Na·Ca, and $HCO_3^-$−Na·Mg type zones, respectively, which suggests complex hydrochemical characteristics of groundwater in the study area [20–22]. The spatial distribution of groundwater facies types in the study area is shown in Figure 4.

It can be seen from Figure 4 that the $HCO_3^-$-Na·Mg facies type of groundwater was observed in a major part of the study area, covering mainly the east-central part of Yongqing Township, Houyi Township, Liujie Township, and the eastern part of Longhuzhuang Township. Whereas $HCO_3^-$−Na, $HCO_3^-$−Mg·Ca, and $HCO_3^-$−Na·Ca facies types were mainly distributed in the southwestern part of the urban area of Yongqing Town, the western part of Yongqing town, and the western part of Longhuzhuang Township, respectively. In addition, the groundwater facies types in the study area are more complex, which suggests influences of human activities on the hydrochemical characteristics of groundwater in the shallow aquifer in the study area [23].

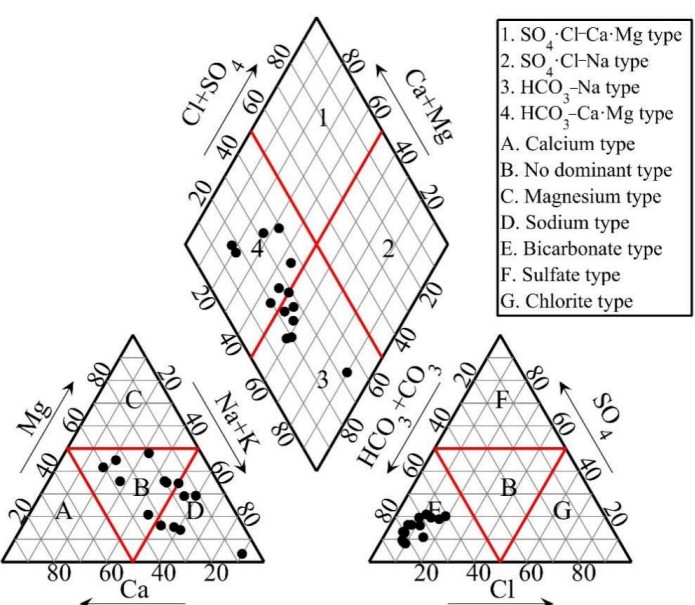

**Figure 3.** Piper diagram of phreatic groundwater in the study area.

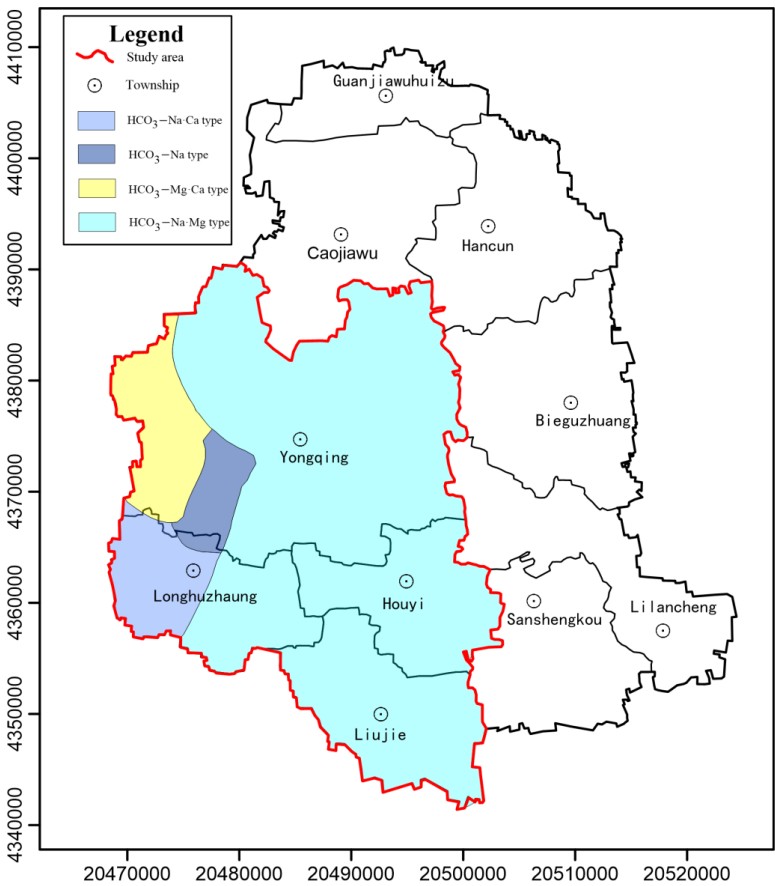

**Figure 4.** Zoning map of groundwater facies types in the study area.

Overall, the relative contents of $HCO_3^-$, $Ca^{2+}$, and $Mg^{2+}$ were higher in fresh and brackish water with low TDS. Water facies types and ion abundances in groundwater were consistent with these hydrochemical results. Yongqing County is located in the alluvial fan plain in front of the Taihang Mountains. The general flow trend of the shallow groundwater is from northwest to southeast runoff. However, the local groundwater flow direction in some areas in Yongqing County is northeast-southwest due to the impact of

the local groundwater landing funnel [24]. Groundwater facies types are mainly influenced by rainfall infiltration and lateral runoff, resulting in low TDS and mineralization of groundwater. The groundwater facies type was mainly represented by the bicarbonate facies type, which suggests the dissolution of calcite, dolomite, and other carbonate minerals from the Taihang Mountains.

*4.3. Analysis of the Evolution Mechanism*

In this study, multivariate statistics and water chemistry analysis methods were carried out to investigate the main factors influencing the hydrochemical characteristics of groundwater Yongqing County [25,26].

4.3.1. PCA (Principal Component Analysis)

Factor analysis is a statistical method used to effectively reduce the number of variables (dimensionality reduction) to minimize the loss of information in the original dataset, thus achieving a comprehensive analysis of data [27]. This statistical method has been widely used in various fields of research, including hydrogeological research. In this study, the factor analysis test was used to investigate the factors influencing the hydrochemical characteristics of groundwater.

It can be seen from the total variance explained that the first 4 principal components explained 87% of the total variance. In addition, the eigen values of the first 4 principal components were all greater than 1. Therefore, the first five principal components were selected. The main influencing factors were assessed to use the factors loading of the principal components (Table 2 and Figure 5). Factor loadings indicate the correlation between variables and principal components (PCs).

**Table 2.** Total variance explained by the factor analysis.

| PCs | Initial Eigenvalue | | |
|---|---|---|---|
| | Total | Variance Percentage | Cumulative Percentage |
| 1 | 9.315 | 51.751 | 51.751 |
| 2 | 3.054 | 16.966 | 68.716 |
| 3 | 1.721 | 9.562 | 78.279 |
| 4 | 1.622 | 9.009 | 87.288 |
| 5 | 0.831 | 4.618 | 91.906 |
| 6 | 0.586 | 3.254 | 95.160 |
| 7 | 0.424 | 2.354 | 97.514 |
| 8 | 0.216 | 1.203 | 98.717 |
| 9 | 0.121 | 0.675 | 99.392 |
| 10 | 0.070 | 0.390 | 99.782 |
| 11 | 0.024 | 0.135 | 99.917 |
| 12 | 0.014 | 0.080 | 99.997 |
| 13 | 0.001 | 0.003 | 100.000 |
| 14 | 0.000 | 0.000 | 100.000 |
| 15 | 0.000 | 0.000 | 100.000 |
| 16 | 0.000 | 0.000 | 100.000 |
| 17 | 0.000 | 0.000 | 100.000 |
| 18 | 0.000 | 0.000 | 100.000 |

According to Tables 2 and 3, it can be seen that the variance contribution of Factor 1 (F1) accounted for 51.751% of the total variance, with strong positive loadings with $HCO_3^-$, $Mg^{2+}$, TH, TDS, and mineralization (M), thus the findings indicate that these five parameters are the most important, representing the dissolution of calcite, dolomite carbonate and salt minerals, and silicate minerals. The results showed that F1 represents the main factors influencing the hydrochemical characteristics of shallow groundwater in the study area. The cumulative variance contribution of factor 2 (F2) was 68.716%. This factor revealed strong positive loadings with $Na^+$ and $F^-$, which indicates that the high

importance of these two parameters in F2. Therefore, it was suggested that F2 was related to the anthropogenic factors represented by $F^-$ concentrations in groundwater. On the other hand, Factor 3 (F3) and Factor 4 (F4) explained smaller proportions of variance of 9.562 and 9.009%, respectively. The result showed strong positive loadings of $NO_2^-$ and As on F3, which indicates the high importance of these parameters in F3. Whereas $NO_3^-$ and $Fe^{2+}$ revealed strong positive and negative loadings on F4, respectively, which indicates the high importance of these two parameters in F4. The results suggested that F3 and F4 are related to industrial and agricultural pollution. The provincial-level industrial Park in Yongqing County, approved by the People's Government of Hebei Province in 2003, may affect the groundwater quality due to industrial wastewater discharges. Moreover, the agricultural area accounts for about 75% of the total area of the study area, which suggests a significant negative impact of agricultural activities on the groundwater quality in Yongqing County.

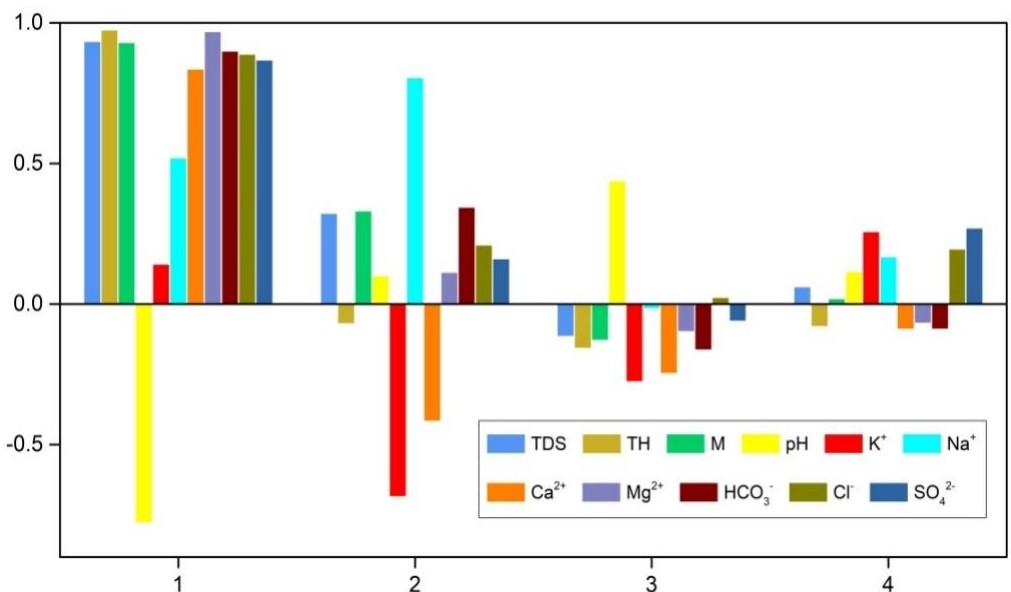

**Figure 5.** Factor loading values between variable and principal components.

**Table 3.** Factor loading values between variables and principal components.

| Chemical Parameters | PCs | | | |
| :---: | :---: | :---: | :---: | :---: |
| | F1 | F2 | F3 | F4 |
| TDS | 0.934 | 0.323 | −0.116 | 0.062 |
| TH | 0.975 | −0.071 | −0.158 | −0.081 |
| M | 0.930 | 0.332 | −0.130 | 0.020 |
| pH | −0.779 | 0.101 | 0.439 | 0.116 |
| $K^+$ | 0.142 | −0.686 | −0.277 | 0.258 |
| $Na^+$ | 0.520 | 0.806 | −0.017 | 0.169 |
| $Ca^{2+}$ | 0.836 | −0.418 | −0.248 | −0.091 |
| $Mg^{2+}$ | 0.970 | 0.114 | −0.099 | −0.069 |
| $HCO_3^-$ | 0.900 | 0.345 | −0.164 | −0.090 |
| $Cl^-$ | 0.889 | 0.211 | 0.024 | 0.196 |
| $SO_4^{2-}$ | 0.868 | 0.161 | −0.062 | 0.271 |
| $F^-$ | 0.326 | 0.822 | −0.232 | 0.154 |
| $NO_3^-$ | 0.184 | 0.193 | 0.019 | 0.743 |
| $NO_2^-$ | −0.212 | −0.112 | 0.883 | 0.173 |
| Mn | 0.883 | −0.067 | 0.041 | −0.292 |
| $Fe^{2+}$ | 0.199 | 0.265 | 0.019 | −0.786 |
| As | −0.039 | 0.097 | 0.946 | −0.125 |
| $SiO_2$ | 0.259 | −0.478 | −0.473 | −0.575 |

Note: Extraction method: principal component analysis; Rotation method: Caesar normalized maximum variance method a; a. The rotation was converged after 6 iterations.

### 4.3.2. Gibbs Diagram Analysis

Gibbs diagrams (Figure 6) have been widely used to reveal the ionic characteristics and determine the sources of the hydrochemical characteristics of river water [28,29]. In addition, they have been commonly used to analyze the hydrochemical characteristics of groundwater. Gibbs diagrams can be used to assess the relationship between TDS and $Na^+/(Na^++Ca^{2+})$ and between TDS and $Cl^-/(Cl^-+HCO_3^-)$ and identify the main sources of ions as well as the main factors influencing the hydrochemical characteristics of groundwater.

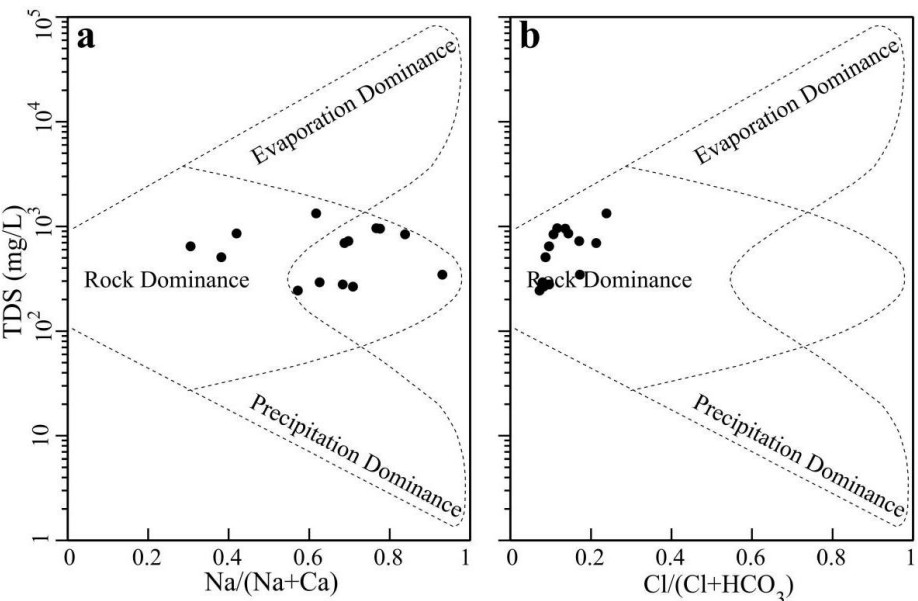

**Figure 6.** Gibbs diagrams, (**a**) TDS versus Na/(Na+Ca), (**b**) TDS versus Cl/(Cl+HCO$_3$).

It can be seen from Figure 6 that all water sampling points fall in the rock dominance control zone, which indicates that that rock dominance was the main influencing factor controlling the hydrochemical characteristics of groundwater in the study area. As shown in Figure 6a, $Na^+$ has a wide distribution and exhibit high proportions in some samples, indicating that sodium ions may be generated by a variety of sources. However, the proportion of $Cl^-$ is small and the distribution is concentrated, indicating that all $Cl^-$ is produced from the similar source or through the same geochemical process (Figure 6b). It should be noted that the Gibbs diagrams revealed only the natural factors influencing the groundwater chemical characteristics, while the human factors were not considered.

### 4.3.3. Inter-Ion Distance Diagram Analysis

According to the results of factor analysis and Gibbs diagram, it is evident that rock dominance and mineral dissolution were the main factors controlling the hydrochemical characteristics of groundwater. Therefore, to further investigate the mineral species derived from rock weathering and dissolution and to verify the results obtained using the principal component analysis, the $HCO_3^-/Na^+$, $Ca^{2+}/Na^+$, $Mg^{2+}/Na^+$, and $Ca^{2+}/Na^+$ ratios were used to distinguish between the influences of different rock and mineral weathering on groundwater components.

The results showed that most of the water sampling points were plotted between silicate weathering and carbonate dissolution zones, which suggests significant influences of the weathering of silicate and carbonate minerals on the hydrochemical characteristics of groundwater (Figures 7 and 8).

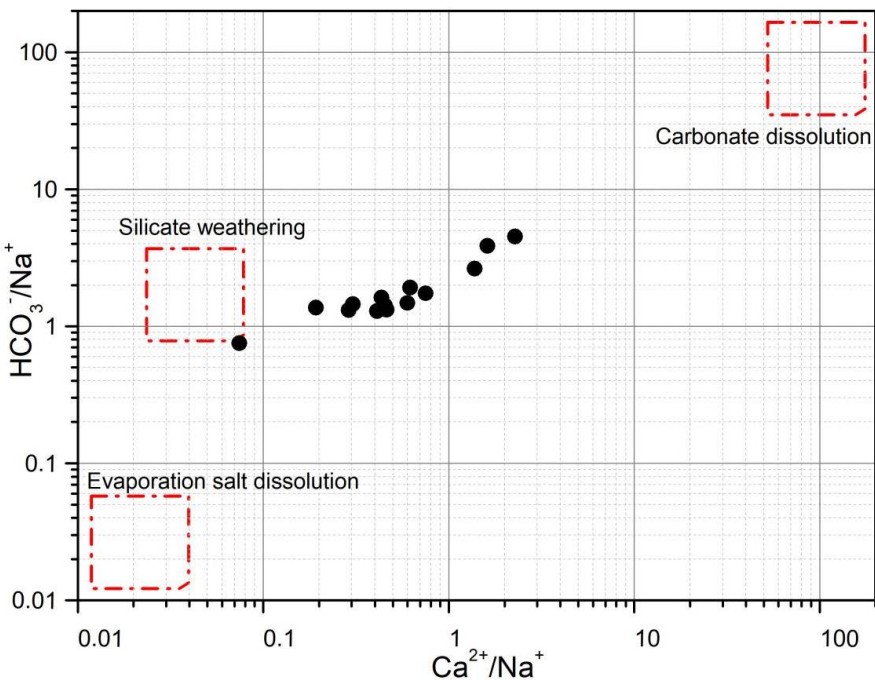

**Figure 7.** Relationship between $Ca^{2+}/Na^+$ and $HCO_3^-/Na^+$ of the phreatic aquifer in Yongqing County.

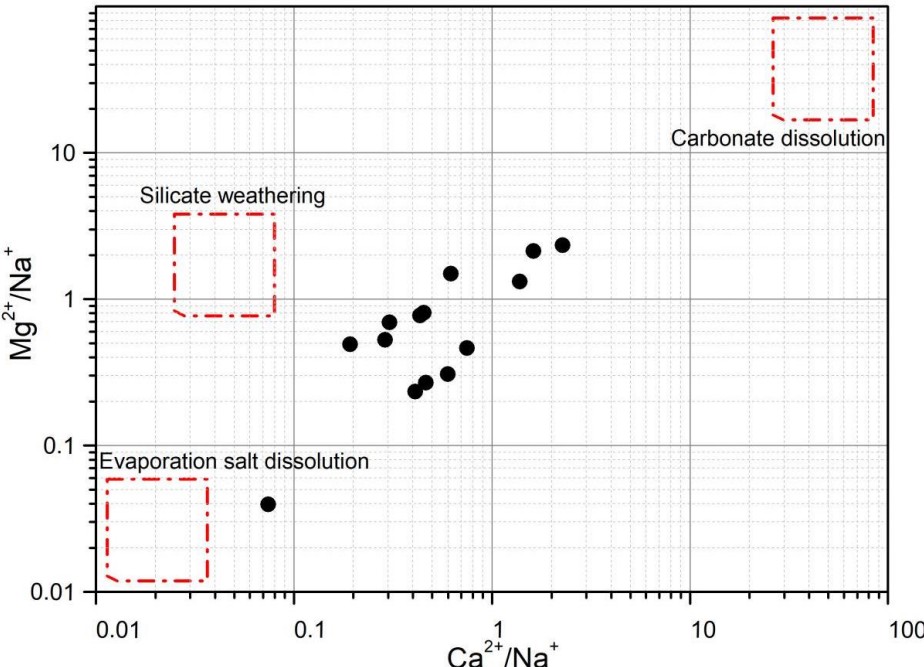

**Figure 8.** Relationship between $Ca^{2+}/Na^+$ and $Mg^{2+}/Na^+$ of phreatic aquifer in Yongqing County.

In order to further check whether the ion exchange process occurred in groundwater, the relationship diagram between $Ca^{2+}+Mg^{2+}-HCO_3^--SO_4^{2-}$ and $Na^++K^+-Cl^-$ were used for discrimination [30]. Water sampling points in the first quadrant of graphs (positive X and Y-coordinate values) suggest that rock salt dissolution is not the source of $Na^+$ and $K^+$, while the dissolution of peritectic minerals is not the source of $Ca^{2+}$ and $Mg^{2+}$ in groundwater, which is explained by the presence of a higher amount of $Na^++K^+$ than $Cl^-$ and lower amount of $Ca^{2+}+Mg^{2+}$ than $HCO_3^--SO_4^{2-}$ (Figure 9). In addition, equal ions concentrations suggest that a cation exchange process occurred in groundwater.

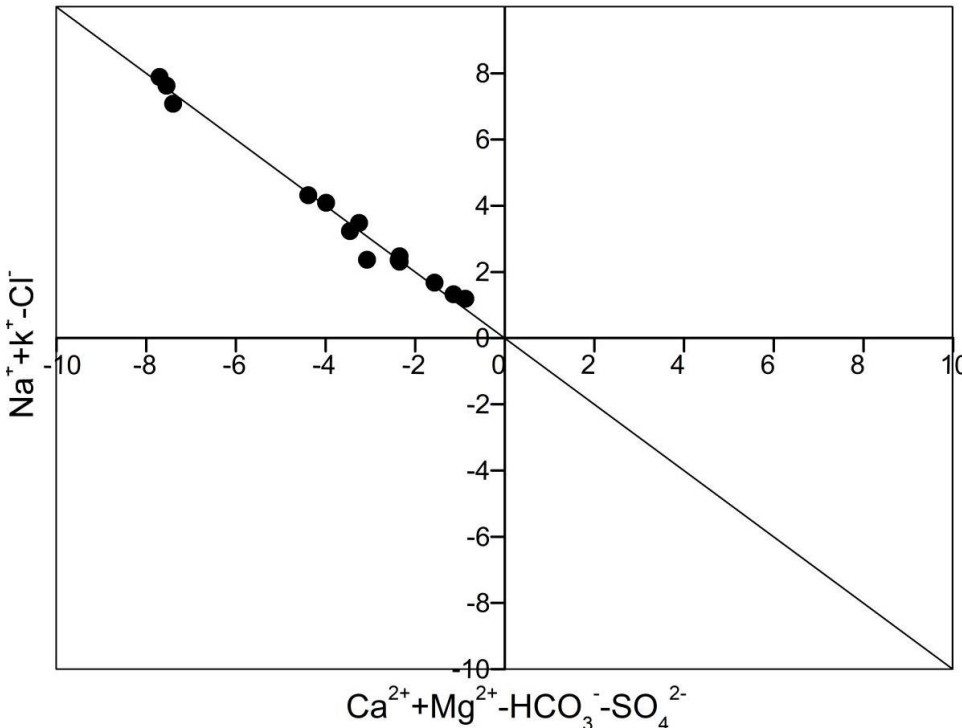

**Figure 9.** Relationship between $Ca^{2+}+Mg^{2+}-HCO_3{}^--SO_4{}^{2-}$ and $Na^++K^+-Cl^-$ ions of phreatic aquifer in Yongqing County.

According to the results obtained, all the water sampling points were plotted on the 1:1 line, indicating that the cation exchange process occurred in the unconfined aquifers of the study area.

*4.4. HydrogeochemicalInverse Simulation*

Yongqing County is located in the pre-mountain alluvial and flood plain of the Tai-hang Mountains. The unconfined aquifers in the study area are mainly recharged from atmospheric precipitation and lateral runoff recharge. In general, the unconfined aquifers of pre-mountain alluvial and floodplain have been controlled by lateral runoff recharge and are characterized by good groundwater quality and a single groundwater facies type, which is inconsistent with the results of this study. In fact, the results revealed complex groundwater facies types of the unconfined aquifers in the study area, the results suggest that numerous factors influencing the hydrochemical characteristics of groundwater quality. Therefore, the PHREEQC software was used in this study to comprehensively assess the hydrogeochemical process, investigate the major complex facies types, and analyze the water-rock interaction in the unconfined aquifers in the study area, providing quantitative analysis results.

4.4.1. Simulation Path Selection

The simulation path was selected in this study based on the geological and hydro-geological settings of the study area to better represent the evolution characteristics of groundwater in the entire study area [31,32]. Since the groundwater flow direction is northeast-southwest due to the influence of the local groundwater funnel, a simulation path was selected along this direction of groundwater flow, taking into the sampling points that are located in this area (Figure 10).

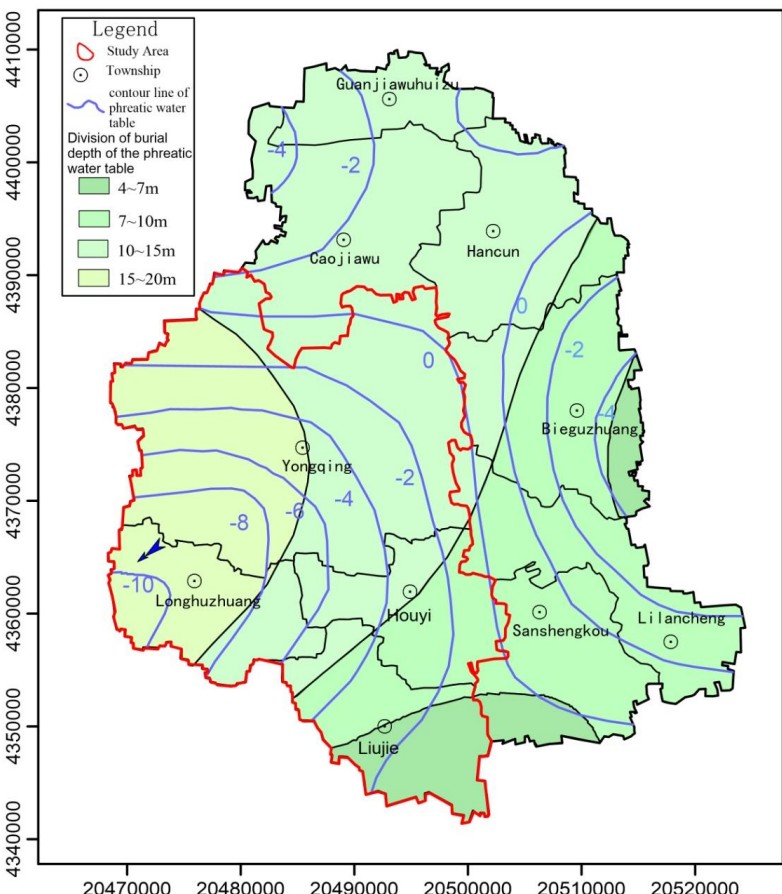

**Figure 10.** Hydrogeochemical inverse simulation path in the study area.

W6 and W12 were selected as the upstream and downstream, respectively. The hydrochemical characteristic results observed at these two sampling points were used as input data in the PHREEQC simulation software to perform inverse hydrogeochemical simulation. In addition, the uncertainty limit was set at 0.05.

### 4.4.2. Selection of Possible Mineral Phases

The hydrogeochemical process of the unconfined aquifers along the groundwater flow direction is closely related to the properties of surrounding rocks. Apparently, the mineral of surrounding rocks can provide insight into the hydrochemical components of groundwater. Therefore, the selection of mineral phases plays a key role in the accuracy of the simulation results.

The study area is located in the North China Plain, where the overlying strata are mainly Quaternary deposits. The unconfined aquifers are of Quaternary loose rock type. $CO_2$, as a possible mineral phase, can be used in the model due to the presence of gas-exchange surface area. In addition, the proportional relationship between ions suggested the occurrence of an exchange cation process in the aquifer, which implies that the reason for considering the ion exchange as a possible mineral phase in the simulation. On the hand, the results of the principal component analysis show that carbonate and silicate dissolutions were the main factors influencing the hydrochemical characteristics of groundwater. Therefore, carbonate minerals (e.g., calcite and dolomite) and aluminosilicate minerals (e.g., potassium feldspar and sodium feldspar) were considered as possible mineral phases in the simulation.

### 4.4.3. Analysis of Simulation Results

The simulation results of the hydrogeochemical processes of groundwater in the study area are shown in Tables 4 and 5.

**Table 4.** Upstream and downstream water saturation indices along the reaction path.

| Mineral | Upstream | Downstream |
|---|---|---|
| Anhydrite | −2.08 | −2.38 |
| Calcite | 0.65 | 0.77 |
| Chalcedony | 0.01 | 0.12 |
| $CO_2(g)$ | −1.80 | −2.15 |
| Dolomite | 1.69 | 1.78 |
| Fluorite | −0.34 | −1.56 |
| Gypsum | −1.78 | −2.08 |
| Halite | −6.42 | −7.48 |
| Hematite | 16.24 | 19.50 |
| Manganite | −5.65 | −3.92 |
| Quartz | 0.44 | 0.55 |
| Sylvite | −8.72 | −8.94 |

**Table 5.** Results of the inverse simulation along the reaction path.

| Phase | Formula | Mole Transfers (mol/L) |
|---|---|---|
| Gypsum | $CaSO_4:2H_2O$ | $-9.29 \times 10^{-4}$ |
| Calcite | $CaCO_3$ | $2.20 \times 10^{-3}$ |
| Calcium exchange | $CaX_2$ | $3.49 \times 10^{-3}$ |
| Sodium exchange | $NaX$ | $-6.34 \times 10^{-3}$ |
| Magnesium exchange | $MgX_2$ | $-3.19 \times 10^{-4}$ |
| Carbon Dioxide(g) | $CO_2(g)$ | $-2.73 \times 10^{-3}$ |
| Albite | $Na_2O \cdot Al_2O_3 \cdot 6SiO_2$ | $2.32 \times 10^{-5}$ |
| K-feldspar | $KAlSi_3O_8$ | $1.93 \times 10^{-5}$ |
| K-mica | $KAl_3Si_3O_{10}(OH)_2$ | $-1.42 \times 10^{-5}$ |
| Fluorite | $CaF_2$ | $-6.49 \times 10^{-5}$ |
| Manganite | $MnO(OH)$ | $3.75 \times 10^{-6}$ |
| Hematite | $Fe_2O_3$ | $1.73 \times 10^{-5}$ |

Note: Positive and negative values indicate dissolution and precipitation, respectively.

The saturation indices of minerals in the upstream and downstream water are reported in Table 4. Positive and negative values indicate oversaturated and unsaturated groundwater samples, respectively. The results showed that calcite, chalcedony, dolomite, hematite, and quartz were all oversaturated. This finding can be explained by the fact that calcite and dolomite are the main minerals of the unconfined aquifers in the study area in the eastern plains in the Taihang Mountain front the remaining minerals revealed negative values, thus indicating unsaturation.

Table 5 shows the molar transfer of possible mineral phases along the reaction path. The results showed precipitation of gypsum, carbon dioxide, K-mica, and fluorite in the unconfined aquifers of the study area. Whereas Calcite, sodium exchange, K-feldspar, manganite, and hematite were in dissolved forms in the unconfined aquifers, with dissolved amounts of $2.20 \times 10^{-5}$, $1.93 \times 10^{-5}$, $3.75 \times 10^{-6}$, and $1.73 \times 10^{-5}$ mol/L, respectively. On the other hand, the cation exchange in the unconfined aquifer was characterized by the exchange of $Ca^{2+}$, $Na^+$, and $Mg^{2+}$, which suggests adsorption of $Na^+$ and $Mg^{2+}$ in groundwater, while $Ca^{2+}$ in the exchange medium was in a dissolved form in groundwater. The amount values of $Na^+$, $Mg^{2+}$, and $Ca^{2+}$ ion exchange were $6.34 \times 10^{-3}$, $3.19 \times 10^{-4}$, and $3.49 \times 10^{-3}$ mol/L, respectively.

The results of the inverse hydrogeochemical simulations performed using the PHREEQC software are consistent with those obtained using the principal components analysis and water chemistry analysis. The main factors influencing the hydrochemical characteristics of

groundwater are the dissolution of carbonate minerals (calcite and dolomite) and silicate minerals (sodium feldspar and potassium feldspar).

## 5. Conclusions

In this study, 14 groundwater samples were collected from the unconfined aquifers of Western Yongqing County to assess the hydrochemical characteristics of groundwater using descriptive, multivariate statistics, and water chemistry analysis methods. The following main conclusions were drawn from this study:

(1) The unconfined aquifers in the western part of Yongqing County revealed weakly alkaline groundwater. In addition, the results suggested fresh and brackish groundwater in the study area. The abundance of cations and anions followed the orders of $Na^+ > Ca^{2+} > Mg^{2+} > K^+ > Fe^{2+}$ and $HCO_3^- > SO_4^{2-} > Cl^- > NO_3^- > F^- > NO_2^-$, respectively.

(2) The relative content of anions and cations in groundwater and Piper's trilinear diagram demonstrated complex hydrochemical facies types of groundwater in the study area. The groundwater facies types were $HCO_3^- - Mg \cdot Ca$, $HCO_3^- - Na$, $HCO_3^- - Na \cdot Ca$, and $HCO_3^- - Na \cdot Mg$.

(3) The main factors influencing the hydrochemical characteristics of groundwater were determined to use the principal component analysis, Gibbs diagrams, and ionic ratios. The results revealed that mineral dissolution, as well as some anthropogenic factors, are the main factors influencing the groundwater chemistry in the study area. In addition, Gibbs diagrams and ionic ratios revealed that silicates and carbonates were the main minerals influencing the hydrochemical characteristics of unconfined aquifers in the study area, followed by the alternating cation adsorption.

(4) Simulation of water-rock interactions in the unconfined aquifers was performed using the PHREEQC software. The simulation results are consistent with those obtained using the principal component analysis, Gibbs diagram, and ionic ratios. According to the obtained results, the dissolution of carbonate minerals (calcite and dolomite) and silicate minerals (sodium feldspar and potassium feldspar) were the main factors influencing the hydrochemical characteristics of the unconfined aquifers in the study area.

In conclusion, although there are anthropogenic activities in some areas of Yongqing County, the groundwater quality of the unconfined aquifers in the western part of Yongqing County is relatively good. Therefore, relevant departments need to strengthen the monitoring and management of groundwater quality to ensure scientific management and sustainable utilization of groundwater in Yongqing County.

**Author Contributions:** Conceptualization, X.B.; methodology, Y.L.; software, J.L.; validation, Y.Z.; formal analysis, X.T.; investigation, X.W.; resources, X.B.; data curation, X.T.; writing—original draft preparation, Y.Z.; writing—review and editing, J.L. All authors have read and agreed to the published version of the manuscript.

**Funding:** This research is funded by Open project of Hebei key laboratory of geological resources and environmental monitoring and protection (JCYKT202101); Natural Science Foundation of Hebei Province of China (D2022403016); Hebei University Science and technology research project (ZD2022119); Science and technology innovation team project of Hebei GEO University (KJCXTD-2021-14); Introduction of foreign intelligence project in Hebei province in 2021 (2021ZLYJ-1); Hebei water conservancy science and technology plan project (2021-45).

**Informed Consent Statement:** Not applicable.

**Data Availability Statement:** The data are all monitored by the Hebei Geological Environment Monitoring Institute, which are true and reliable.

**Acknowledgments:** The authors acknowledge the support provided by Hebei GEO University; Hebei Key Laboratory of Environment Monitoring and Protection of Geological Resources, Hebei Geological Environment Monitoring Institute, Shijiazhuang; Hebei Institute of hydrogeology and engineering geology.

**Conflicts of Interest:** The authors declare no conflict of interest.

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
