# Peer review of "Assessment of the Hydrochemical Characteristics and Formation Mechanisms of Groundwater in A Typical Alluvial-Proluvial Plain in China: An Example from Western Yongqing County"

_water, doi:10.3390/w14152395_

Round 1
Reviewer 1 Report
I have read the present work attentively.
I have a mile of suggestions.
1. figure 1. it is placed correctly but because of the description of the location, maybe it is worth referring to this figure already in chapter 2.1? In addition, in fig. 1. lacks neighboring countries and in the text (in the introduction) there is a reference to Korea, in view of this it would be worthwhile to outline where specifically are the border areas of this country. A reader from outside the East Asian region may have a problem with this.
2. it would be worthwhile to better explain the hydrological contact of the layers in section 2.3. it is necessary to specify whether there are isolation factors, if not, what is the estimated flow of water to the various layers, etc. This is important in determining further argument.
3. Figure 6 has two graphs (a) and (b), please explain both in the caption.
One more suggestion: in indicating the sources of groundwater mixing, I think it is worth thinking about isotopic analysis as well. This would provide a lot of conclusive data.
In general, I consider the text to be written correctly.
Author Response
Thank you for your suggestion, I have made the corresponding changes as follows.
1、Figure 1 has been marked in 2.1, and the geographical location of the Nakdong River basin, Busan, Korea,relative to China has been explained in the introduction, and Korea is mentioned in the introduction as an example to explain the current approach to groundwater studies.
2、There is no isolation factor between aquifers, but the water flow in each layer is not quite the same due to the heterogeneity of aquifers.
3、The two figures (a) and (b) of Figure 6 have been explained in detail in the text.
4、This is a very good suggestion, but in this study we do not measure isotope data, and we will consider it properly in future studies.
Reviewer 2 Report
Major comments:
1. This manuscript (ms) is required a proofreading and editing by a native English speaker before being resubmitted for consideration.
2. This ms includes numerous "indeed". The ms would sounds better by simply removing these "indeed".
3. The authors used numerous "indicating", "suggesting", "representing" and "explaining" in the ms to express facts or conclusions. Such expressions should be improved for reconsideration. Following lists a few better expressions recommended by the reviewer
(1) The findings indicate that
(2) The results suggest that
(3) which implies that
4. Lines 162-163, "Finally, the status of the samples is checked regularly. Finally, the status of the samples is checked regularly". One of the sentence is redundant.
5. Lines 386-387, "The inverse hydrogeochemical simulation was carried out using two sampling points (W6 and W12). The uncertainty limit value was set at 0.05." These redundant statements should be removed.
Author Response
Thank you for your comments. I have made the appropriate changes based on these comments, which are as follows:
- This manuscript has been proofread and revised by native English speaking professionals.
- Most of the words "indeed" have been deleted. In addition, some of the "indeed" was replaced by "in fact", "Apparently" and other near-synonyms.
- I have replaced "indicating", "suggesting", "presenting" and " explaining" with the following expressions:
(1) The findings indicate that/which indicates
(2) The results suggest that/which suggests
(3) which implies that
- the sentence "Finally, the status of the samples is checked regularly." has been deleted.
- "The inverse hydrogeochemical simulation was carried out using two sampling points (W6 and W12). The uncertainty limit value was set at 0.05." has been deleted.